# Metformin and the Risk of Chronic Urticaria in Patients with Type 2 Diabetes

**DOI:** 10.3390/ijerph191711045

**Published:** 2022-09-03

**Authors:** Fu-Shun Yen, Chih-Cheng Hsu, Kai-Chieh Hu, Yu-Tung Hung, Chung Y. Hsu, James Cheng-Chung Wei, Chii-Min Hwu

**Affiliations:** 1Dr. Yen’s Clinic, No. 15, Shanying Road, Gueishan District, Taoyuan 33354, Taiwan; 2Institute of Population Health Sciences, National Health Research Institutes, 35 Keyan Road, Zhunan, Miaoli County 35053, Taiwan; 3Department of Health Services Administration, China Medical University, No. 91, Hsueh-Shih Road, Taichung 40402, Taiwan; 4Department of Family Medicine, Min-Sheng General Hospital, 168 ChingKuo Road, Taoyuan 33044, Taiwan; 5National Center for Geriatrics and Welfare Research, National Health Research Institutes, 35 Keyan Road, Zhunan, Miaoli County 35053, Taiwan; 6Management Office for Health Data, China Medical University Hospital, 3F, No. 373-2, Jianxing Road, Taichung 40459, Taiwan; 7College of Medicine, China Medical University, No. 91, Xueshi Road, Taichung 40202, Taiwan; 8Graduate Institute of Biomedical Sciences, China Medical University, No. 91, Hsueh-Shih Road, Taichung 40402, Taiwan; 9Department of Allergy, Immunology & Rheumatology, Chung Shan Medical University Hospital, Taichung 40203, Taiwan; 10Institute of Medicine, Chung Shan Medical University, Taichung 40203, Taiwan; 11Graduate Institute of Integrated Medicine, China Medical University, Taichung 40202, Taiwan; 12Faculty of Medicine, School of Medicine, National Yang-Ming Chiao Tung University, No. 155, Sec. 2, Linong Street, Taipei 11221, Taiwan; 13Section of Endocrinology and Metabolism, Department of Medicine, Taipei Veterans General Hospital, No. 201, Sec. 2, Shipai Road, Beitou District, Taipei 11217, Taiwan

**Keywords:** type 2 diabetes, metformin, chronic urticaria, development, hospitalization

## Abstract

We conducted this study to determine the effect of metformin use on the risk of new-onset chronic urticaria in patients with type 2 diabetes (T2D). In total, 24,987 pairs of metformin users and nonusers were identified with propensity score-matching from Taiwan’s National Health Insurance Research Database from 1 January 2000, to 31 December 2017. Multivariable Cox proportional hazards models were used to compare the risks of chronic urticaria development, severe chronic urticaria, and hospitalization for chronic urticaria between metformin users and nonusers. Compared with metformin nonuse, the aHRs (95% CI) for metformin use in chronic urticaria development, severe chronic urticaria, and hospitalization for chronic urticaria were 1.56 (1.39–1.74), 0.40 (0.12–1.30), and 1.45 (0.82–2.56), respectively. The cumulative incidence of chronic urticaria development was significantly higher in metformin users than in nonusers (*p* < 0.0001). A longer average cumulative duration of metformin use was associated with higher risks of new-onset and hospitalization for chronic urticaria than metformin nonuse. This nationwide cohort study showed that metformin use was associated with a significantly higher risk of chronic urticaria development. A longer average cumulative duration of metformin use was associated with a higher risk of outcomes. More prospective studies are needed to verify our results.

## 1. Introduction

Many people experience itchy, scratchy, and inflamed skin, which leads to eczema [1]. The most common type of eczema is atopic dermatitis, which occurs mainly in children [2]. Chronic urticaria is considered if the condition presents as relapsing and distressing wheals, with or without angioedema, and lasts for more than 6 weeks [3]. Chronic urticaria often occurs in adults, women, or people with psychological stress, and food, infections, and drugs are the risk factors for chronic urticaria [3]. The lifetime prevalence of chronic urticaria is about 9% [3], and about 15–23% of Americans have chronic urticaria [4]. Reports show that diabetes mellitus has a slightly higher risk of chronic urticaria [5]. Mast cell activation with subsequent degranulation and histamine release is the main cause of wheals in patients with chronic urticaria. Some patients also have functional immunoglobulin G (IgG) autoantibodies against high-affinity IgE receptors (FcεRIα) or IgE. Chronic urticaria seems to be an autoimmune, mast-cell disease with complicated mechanisms [3,4].

Metformin, approved for use by the Food and Drug Administration in 1994, became the first-line antidiabetic drug as the UK Prospective Diabetes Study demonstrated that metformin could significantly reduce the risks of diabetes-related clinical endpoints and all-cause mortality in patients with type 2 diabetes (T2D) in 1997 [6]. In addition to its glucose-lowering property, metformin also has anti-proliferative, anti-aging, and immunomodulating effects [6,7]. Animal studies have demonstrated that metformin can alter the levels of cytokines [7], the ratio of T regulatory (Treg) and T helper 17 (Th17) cells [7,8], and the activation of mast cells [9]. Metformin may alter the patient’s immune and allergic responses. Some studies have shown that metformin might reduce [10,11] or increase [12] the development of asthma. However, no study has explored the relationship between metformin and chronic urticaria risk. Therefore, we conducted this study to determine whether metformin could affect the development of chronic urticaria in patients with T2D.

## 2. Materials and Methods

### 2.1. Study Population and Design

We identified patients with newly diagnosed T2D from the National Health Insurance Research Database (NHIRD) between 1 January 2000, and 31 December 2017. The data source of NHIRD is described in our previous study [12]. This study was approved by the Research Ethics Committee of China Medical University and Hospital [CMUH109-REC2-031(CR-2)]. The trackable information of patients and care providers was de-identified and encrypted before release to avoid personal information leakage. Informed consent was waived by the Research Ethics Committee. Definitions of diseases based on the ICD-9/10 Clinical Modification (CM) coding system are listed in Appendix A. The T2D patients must have at least one hospitalization or 3 outpatient records. The algorithm for using ICD-9/10 codes to define T2D was validated by the previous study in Taiwan, with an accuracy of 74.6% [13]. Patients who received metformin treatment for ≥28 days were defined as study cases and those who never received metformin as control cases. The first date of metformin use after T2D diagnosis was defined as the index date, and the index date of the control group was taken with the same period from T2Ddiagnosis to the first date of metformin use in the study group. Patients were excluded for the following reasons (Figure 1): (1) age below 20 or above 80 years, (2) missing data on sex or age, (3) diagnoses of type 1 diabetes, hepatic failure, or dialysis, (4) at least 2 outpatient visits for primary or secondary vasculitis and allergic purpura within 3 months before or after the index date to exclude patients with urticarial vasculitis [3], (5) diagnosis of T2D before 1 January 2000, to exclude previous T2D cases.

### 2.2. Procedures

Some clinically relevant variables compared and matched between metformin users and nonusers included age, gender, smoking status (ICD codings), obesity (ICD codings), alcohol-related disorders, hypertension, dyslipidemia, coronary artery disease, stroke, heart failure, peripheral arterial disease, chronic kidney disease, chronic obstructive pulmonary disease (COPD), systemic lupus erythematosus, rheumatoid arthritis, liver cirrhosis, cancers, depression, psychosis, and dementia diagnosed before the index date. Comorbidities mentioned above were identified for at least 3 outpatient visits or one hospitalization. We recorded the number of oral antidiabetic drugs (<2, 2–3, >3), the examination of HbA1c >2 times per year during the follow-up time, and the use of glucagon-like peptide-1 receptor agonists (GLP-1RAs), insulins, non-steroidal anti-inflammatory drugs (NSAIDs), statins, and aspirin. We calculated the Charlson Comorbidity Index (CCI) and Diabetes Complication Severity Index (DCSI) scores [14,15] to evaluate the complications of T2D. They were compared and matched between two groups as well.

### 2.3. Main Outcomes

We investigated and compared the outcomes of new-onset chronic urticaria, severe chronic urticaria, and hospitalization for chronic urticaria during the follow-up time. Patients with new-onset chronic urticaria were defined as having ≥ two outpatient visits, or one hospitalization, and receiving H1 or H2 antihistamines (at least 6 weeks apart). Severe chronic urticaria was defined in patients diagnosed with chronic urticaria and receiving third-line treatments, such as azathioprine, cyclosporine, cyclophosphamide, mycophenolic acid, methotrexate, omalizumab, or interferon [16]. Hospitalization for chronic urticaria was defined for patients with one hospitalization record for chronic urticaria. The incidence rates of outcomes were calculated by the timeline of 1000 person-years of follow-up time. To assess the outcomes of interest, we censored patients till the day of respective outcomes, mortality, or at the end of follow-up time on 31 December 2017.

### 2.4. Statistical Analysis

Propensity-score matching was used to balance the critical variables between metformin users and nonusers [17]. We estimated the propensity score for each patient using non-parsimonious multivariable logistic regression with metformin use as a dependent variable and 40 clinically related variables, including age, sex, smoking, obesity, comorbidities, CCI, DCSI scores, medications, and HbA1C check-up >2 times per year, as independent variables (Table 1). The control group (metformin no-use) was matched without replacement. We used the nearest-neighbor algorithm to match pairs and assumed the standardized mean difference (SMD) of less than 0.1 to be negligible between the study and control groups.

Crude and multivariable-adjusted Cox proportional hazards models were applied to compare the outcomes between the study and control groups. The results were presented as hazard ratios (HRs) and 95% confidence intervals (CIs). The Kaplan–Meier method was used to describe the cumulative incidence of chronic urticaria development between metformin nonusers and users during the follow-up period. Log-rank tests were used for examining the difference between the cumulative curves. We also evaluated the association between the average cumulative duration (cumulative days of metformin use/follow-up years) of metformin use and the risk of new-onset chronic urticaria, severe chronic urticaria, and hospitalization for chronic urticaria compared with metformin nonuse.

SAS (version 9.4; SAS Institute, Cary, NC, USA) was used for the statistical analysis, and a two-tailed *p*-value of less than 0.05 was considered significant.

## 3. Results

### 3.1. Participants

We identified 276,415 patients with newly diagnosed T2D from the NHIRD from 1 January 2000 to 31 December 2017. Of them, 206,046 were metformin users, and 70,369 were nonusers (Figure 1). After excluding ineligible patients, 1:1 propensity score matching was used to obtain 24,987 pairs of metformin users and nonusers. Variables, such as sex, age, smoking, obesity, alcohol-related disorders, comorbidities, CCI, DCSI scores, medications, the duration of T2D, and > 2 times HBA1c check-ups per year, were well-matched between the study and control groups with all SMD < 0.1 (Table 1). In the matched cohorts, 48.457% of patients were female, and the mean (SD) age was 59.13 (12.40) years. The mean follow-up time for metformin users and nonusers was 6.50 (3.98) and 5.78 (4.16) years, respectively. The mean follow-up time for metformin users was longer but not significant than that for nonusers.

### 3.2. Main Outcomes

In the matched cohorts (Table 2), 819 (3.28%) metformin users and 496 (1.99%) nonusers developed chronic urticaria during the follow-up time (incidence rate: 5.17 versus 3.49 per 1000 person-years). The multivariable model showed that metformin users had a significantly higher risk of new-onset chronic urticaria (aHR = 1.56, 95% CI = 1.39–1.74, *p* < 0.0001) than nonusers (Table 2). The multivariable-adjusted analyses also showed that metformin users had a non-significant lower risk of severe chronic urticaria (aHR = 0.4, 95% CI = 0.12–1.30, *p* = 0.1264) and a non-significant higher risk of hospitalization for chronic urticaria (aHR = 1.45, 95% CI = 0.82–2.56, *p* = 0.2009) than nonusers (Table 2). Patients with female sex, coronary artery disease, chronic obstructive pulmonary disease, using sulfonylurea, insulin, and non-steroidal anti-inflammatory drugs had a higher risk of incident chronic urticaria; however, those with psychosis, CCI score ≥ 2, and statin use had a lower risk of incident chronic urticaria (Appendix A). We have observed the effect of metformin use versus no-use on the risk of chronic urticaria among different groups of sex, coronary artery disease, chronic obstructive pulmonary disease, rheumatoid arthritis, psychosis, CCI, sulfonylureas, insulin, non-steroidal anti-inflammatory drugs, and statins. The subgroup analysis showed that metformin use was associated with a higher risk of chronic urticaria than no use in all groups, with significant interaction in the group of sulfonylureas (Appendix A).

The Kaplan-Meier model showed that the cumulative incidence of new-onset chronic urticaria was significantly higher in metformin users than nonusers (Log-rank test *p*-value < 0.0001; Figure 2).

### 3.3. Cumulative Duration of Metformin Use

The average cumulative duration of 32–88 days/years of metformin use was associated with a significantly higher risk of new-onset chronic urticaria [aHR 1.21 (1.01, 1.44)], and the average cumulative duration of > 145 days/years of metformin use was associated with a significantly higher risk of new-onset chronic urticaria [aHR 3.02 (2.62, 3.48)] and hospitalization for chronic urticaria [aHR 2.27 (1.09, 4.74)] than metformin no-use (Table 3).

## 4. Discussion

This study showed that metformin use was associated with a significantly higher risk of chronic urticaria development. A longer average cumulative duration of metformin use was associated with a higher risk of chronic urticaria development than metformin no-use. A cross-sectional study by Shalom et al. showed that diabetes was associated with a significantly higher risk of chronic urticaria (OR = 1.08, 95% CI 1.01–1.15) [5]. 

Few studies have investigated the relationship between metformin and immune disorders. Chronic urticaria is a disease with allergic and autoimmune characteristics; hence, we perused the literature on the occurrence of allergic and autoimmune diseases with metformin use. Rayner et al. used the UK primary care dataset to perform a cohort study and compare the risk of new-onset asthma among various antidiabetic drugs. Results showed that metformin use was associated with a reduced risk of asthma development [10]. Chen et al. conducted a nested case-control study in Taiwan to compare the asthma risk among different antidiabetic drugs and showed that metformin was associated with a lower risk of incident asthma [11]. Two cohort studies from Taiwan and the United States showed that metformin users had a lower risk of asthma-related hospitalization and exacerbation than nonusers [18,19]. However, our previous cohort study in Taiwan, which included several well-matched critical confounding factors, showed that metformin use was associated with higher risks of asthma development, hospitalization, and exacerbation in patients with T2D [12]. Naffaa et al. conducted a population-based cohort study and showed that adherence to metformin was associated with a lower risk of rheumatoid arthritis development in women [20]. Brauchli et al. used the United Kingdom-based General Practice Research Database for a case-control study and revealed that long-term metformin use was associated with a reduced risk of incident psoriasis [21]. Wu et al. conducted a nationwide nested case-control study in Taiwan and showed that frequent metformin users exhibited a lower risk of psoriasis development than nonusers [22]. However, some case reports show bullous pemphigoid, lichen planus, leukocytoclastic vasculitis, and psoriasiform drug eruptions developing after metformin use in patients with T2D [23] or polycystic ovary syndrome [24]. 

We conducted a nationwide population-based cohort study with propensity-score matching to balance metformin users and nonusers. Results showed that metformin users had a significantly higher risk of new-onset chronic urticaria than nonusers, and a longer average cumulative duration of metformin use was associated with a higher risk of chronic urticaria than metformin no-use. To the best of our knowledge, our study is the first to explore the relationship between metformin and chronic urticaria. Although our study showed that metformin could bring negative results, we included well-matched 40 crucial variables, such as sex, age, obesity, smoking, alcohol-related disorders, comorbidities, and medications, to achieve maximal balance in the condition between metformin users and nonusers. We matched the number of oral antidiabetic drugs and duration of T2D to balance the severity of T2D; we matched the frequency of HbA1c check-ups per year to balance the health utility between the study and control groups and increase their comparability. Our study also revealed that patients with female sex, coronary artery disease, chronic obstructive pulmonary disease (higher disease burden), using sulfonylurea, insulin, and non-steroidal anti-inflammatory drugs (more medications) were associated with a higher risk of chronic urticaria development, which was consistent with previous studies [3,4,5]. The incidence rates of severe chronic urticaria and hospitalization for chronic urticaria may be too low to demonstrate a statistically significant difference between the study and control groups. Our findings are not consistent with the studies by Naffaa and Brauchli et al. [21,22]. Their studies showed that metformin might reduce the risk of autoimmune diseases. The reason for the different findings among these studies may be due to the different outcomes observed, and the different populations and ethnic groups assessed. It may also suggest that though chronic urticaria has both autoimmune and allergic reactions, the role of allergic reactions may be more dominant. However, we have performed a subgroup analysis to determine the effect of metformin use on the risk of chronic urticaria in patients with or without asthma (Appendix A). It showed that metformin use was associated with a significantly higher risk of chronic urticaria in patients with or without asthma, and the interaction p value was no significant difference.

Metformin decreases pro-inflammatory cytokines, increases anti-inflammatory cytokines, regulates the differentiation of regulatory T cells and memory T cells, changes the ratios of Th1/Th2 and Treg /Th17 cells, decreases B cell-intrinsic inflammation, and stabilizes mast cells to modulate immune response via the adenosine monophosphate-activated protein kinase (AMPK)-dependent or independent pathways in animal and human studies [8,9,25]. However, Saenwongsa et al. showed that metformin use was associated with decreased interferon (IFN)-αexpression in patients with T2D, which may decrease the selection of germinal centers and affinity maturation of antibodies [26]. Shore et al. demonstrated that metformin could not attenuate innate airway hyperresponsiveness in *db/db* mice with ozone (O3) exposure [27]. More immunological studies are needed to clarify the effects of metformin on different tissues, gut flora, mast cells, basophils, T cells, B cells, cytokines, and immunoglobulins to understand the exact mechanisms of metformin on the development of chronic urticaria.

This research has some limitations. First, this administrative dataset lacked detailed information on family history of allergic or autoimmune diseases, dietary habits, exercise, occupational exposure, and air pollution, which may relate to the development of chronic urticaria. The NHIRD lacks complete data on biochemical tests, hemoglobin A1C, allergens, and immune functional assays, precluding evaluation of T2D status and patients’ immune function. However, we used some crucial variables, such as gender, age, obesity, smoking, alcohol-related disorders, comorbidities, CCI, and medications, as proxy variables to balance the disease burden between the study and control groups. We used the item and number of oral antidiabetic drugs, insulin, DCSI scores, and duration of T2D as proxy variables to balance the severity of T2D and the number of HbA1c check-ups per year as a proxy variable to evaluate the healthy user bias between metformin users and nonusers and increase their comparability. Though we have done strict statistical confounding procedures, the lack of history and test results may lead to an overestimation of the association in our analysis. Second, physicians may not prescribe metformin to patients with severe renal or liver failure to avoid lactic acidosis. We excluded patients diagnosed with dialysis or hepatic failure from the study to decrease the possibility of confounding by indication. Metformin, as the first-line therapy of T2D, has recently received emphasis. However, some physicians in Taiwan prefer to use sulfonylureas to treat T2D for quick lowering of blood glucose previously. Some patients have abdominal discomfort after metformin use and wish to switch to other antidiabetic drugs. This dataset could not assess the preference of prescription by doctors, medication use by patients, and patient compliance with antidiabetic drugs. Fourth, this study mainly investigated Chinese patients; hence, the results may not be generalizable to other races. Finally, a retrospective cohort study always has some unknown or unobserved bias, and a randomized control study is warranted to confirm our results. 

## 5. Conclusions

This study showed that metformin use could increase the risk of new-onset chronic urticaria in patients with T2D. If patients complain of remitting and relapsing wheals or angioedema after metformin use, doctors may need to evaluate the possibility of chronic urticaria. More prospective studies are needed to substantiate our results, and more basic research to explore the mechanisms of metformin use and chronic urticaria development.

## Figures and Tables

**Figure 1 ijerph-19-11045-f001:**
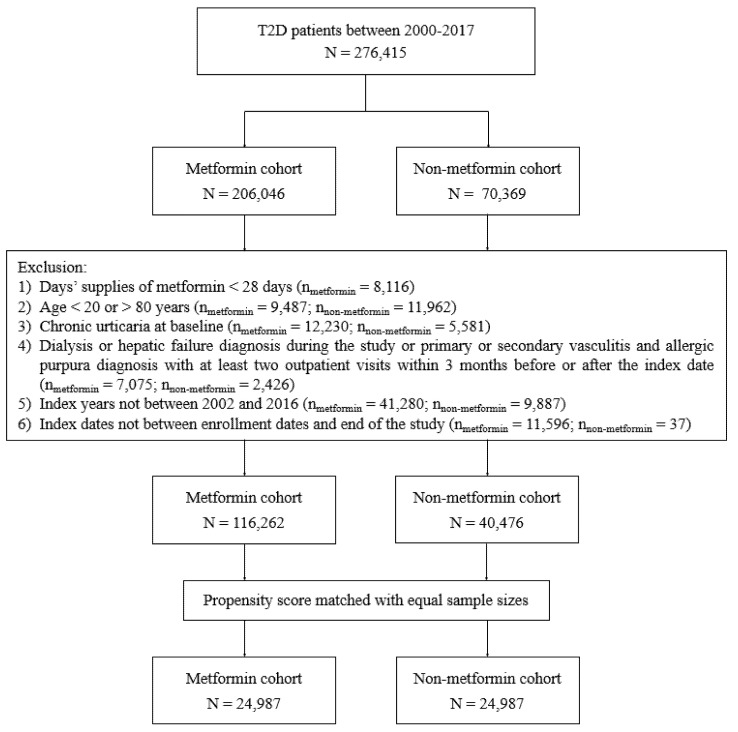
Flow chart of study population selection.

**Figure 2 ijerph-19-11045-f002:**
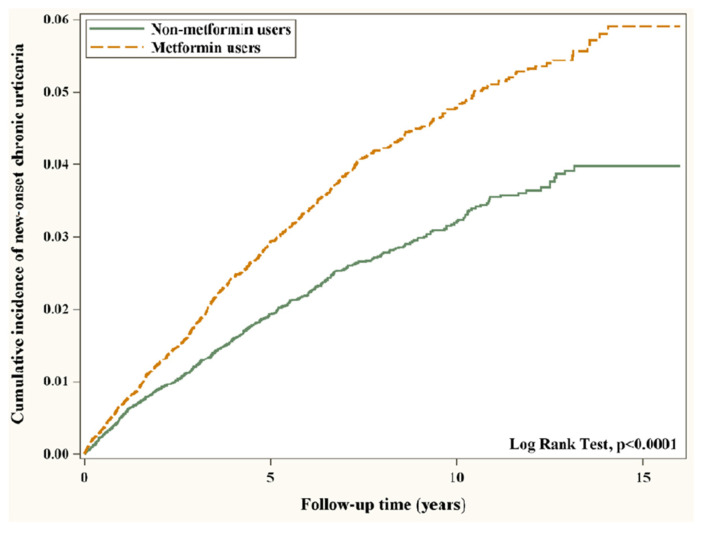
Cumulative incidence of new-onset chronic urticaria in T2D patients with and without metformin. Abbreviation: T2D, type 2 diabetes.

**Table 1 ijerph-19-11045-t001:** Demographic characteristics, comorbidities, and medications in T2D patients with and without metformin.

Variable	Pre-Matched			Post-Matched		
	Non-Metformin Users	Metformin Users	SMD ^§^	Non-Metformin Users	Metformin Users	SMD ^§^
	n (%)	n (%)		n (%)	n (%)	
All	40,476	116,262	0.2784	24,987	24,987	0.0020
Sex			0.1221			0.0199
Female	20,381 (50.35)	51,467 (44.27)		11,981 (47.95)	12,230 (48.95)	
Male	20,095 (49.65)	64,795 (55.73)		13,006 (52.05)	12,757 (51.05)	
Age group (year)						
20–39	3029 (7.48)	9948 (8.56)	0.0395	1949 (7.80)	1736 (6.95)	0.0326
40–59	15,986 (39.50)	59,541 (51.21)	0.2370	10,462 (41.87)	10,185 (40.76)	0.0225
60+	21,461 (53.02)	46,773 (40.23)	0.2585	12,576 (50.33)	13,066 (52.29)	0.0392
Age (year, mean ± SD)	59.51 ± 12.50	56.27 ± 11.86	0.2665	58.85 ± 12.50	59.41 ± 12.30	0.0453
Comorbidities						
Obesity			0.0008			0.0098
No	39,936 (98.67)	114,700 (98.66)		24,649 (98.65)	24,620 (98.53)	
Yes	540 (1.33)	1562 (1.34)		338 (1.35)	367 (1.47)	
Smoking status			0.0297			0.0015
No	39,724 (98.14)	114,544 (98.52)		24,561 (98.30)	24,556 (98.28)	
Yes	752 (1.86)	1718 (1.48)		426 (1.70)	431 (1.72)	
Alcohol-related disorders			0.0440			0.0029
No	39,188 (96.82)	113,409 (97.55)		24,189 (96.81)	24,176 (96.75)	
Yes	1288 (3.18)	2853 (2.45)		798 (3.19)	811 (3.25)	
Hypertension			0.0713			0.0542
No	16,440 (40.62)	51,314 (44.14)		9934 (39.76)	9276 (37.12)	
Yes	24,036 (59.38)	64,948 (55.86)		15,053 (60.24)	15,711 (62.88)	
Dyslipidemia			0.2041			0.0546
No	17,065 (42.16)	60,802 (52.30)		11,237 (44.97)	10,561 (42.27)	
Yes	23,411 (57.84)	55,460 (47.70)		13,750 (55.03)	14,426 (57.73)	
Coronary artery disease			0.2014			0.0378
No	28,037 (69.27)	90,790 (78.09)		17,813 (71.29)	17,382 (69.56)	
Yes	12,439 (30.73)	25,472 (21.91)		7174 (28.71)	7605 (30.44)	
Stroke			0.2054			0.0332
No	33,211 (82.05)	103,727 (89.22)		20,946 (83.83)	20,636 (82.59)	
Yes	7265 (17.95)	12,535 (10.78)		4041 (16.17)	4351 (17.41)	
Heart failure			0.1284			0.0111
No	37,635 (92.98)	111,513 (95.92)		23,350 (93.45)	23,281 (93.17)	
Yes	2841 (7.02)	4749 (4.08)		1637 (6.55)	1706 (6.83)	
Peripheral arterial disease			0.0841			0.0041
No	39,085 (96.56)	113,864 (97.94)		24,217 (96.92)	24,199 (96.85)	
Yes	1391 (3.44)	2398 (2.06)		770 (3.08)	788 (3.15)	
Chronic kidney disease			0.2572			0.0019
No	36,554 (90.31)	112,328 (96.62)		23,196 (92.83)	23,208 (92.88)	
Yes	3922 (9.69)	3934 (3.38)		1791 (7.17)	1779 (7.12)	
Chronic obstructive pulmonary disease			0.1964			0.0156
No	32,434 (80.13)	101,549 (87.34)		20,492 (82.01)	20,341 (81.41)	
Yes	8042 (19.87)	14,713 (12.66)		4495 (17.99)	4646 (18.59)	
Rheumatoid arthritis			0.0787			0.0009
No	39,455 (97.48)	114,602 (98.57)		24,480 (97.97)	24,477 (97.96)	
Yes	1021 (2.52)	1660 (1.43)		507 (2.03)	510 (2.04)	
Systemic lupus erythematous			0.0547			0.0077
No	40,310 (99.59)	116,114 (99.87)		24,914 (99.71)	24,924 (99.75)	
Yes	166 (0.41)	148 (0.13)		73 (0.29)	63 (0.25)	
Liver cirrhosis			0.0934			0.0130
No	39,274 (97.03)	114,427 (98.42)		24,360 (97.49)	24,308 (97.28)	
Yes	1202 (2.97)	1835 (1.58)		627 (2.51)	679 (2.72)	
Cancers			0.1492			0.0137
No	37,796 (93.38)	112,335 (96.62)		23,659 (94.69)	23,581 (94.37)	
Yes	2680 (6.62)	3927 (3.38)		1328 (5.31)	1406 (5.63)	
Psychosis			0.0434			0.0059
No	39,507 (97.61)	114,199 (98.23)		24,438 (97.80)	24,416 (97.71)	
Yes	969 (2.39)	2063 (1.77)		549 (2.20)	571 (2.29)	
Depression			0.1390			0.0070
No	37,855 (93.52)	112,237 (96.54)		23,685 (94.79)	23,646 (94.63)	
Yes	2621 (6.48)	4025 (3.46)		1302 (5.21)	1341 (5.37)	
Dementia			0.0886			0.0073
No	40,185 (99.28)	116,099 (99.86)		24,871 (99.54)	24,883 (99.58)	
Yes	291 (0.72)	163 (0.14)		116 (0.46)	104 (0.42)	
Charlson Comorbidity Index						
0	26,627 (65.78)	92,391 (79.47)	0.3106	17,269 (69.11)	16,821 (67.32)	0.0385
1	5201 (12.85)	12,028 (10.35)	0.0783	3193 (12.78)	3412 (13.66)	0.0259
2+	8648 (21.37)	11,843 (10.19)	0.3104	4525 (18.11)	4754 (19.03)	0.0236
Diabetes Complications Severity Index						
0	12,926 (31.93)	54,792 (47.13)	0.3146	8964 (35.87)	8036 (32.16)	0.0785
1	7530 (18.60)	21,500 (18.49)	0.0029	4692 (18.78)	4936 (19.75)	0.0248
2+	20,020 (49.46)	39,970 (34.38)	0.3093	11,331 (45.35)	12,015 (48.09)	0.0549
Medications						
Numbers of oral antidiabetic drugs						
<2	39,780 (98.28)	111,718 (96.09)	0.1327	24,393 (97.62)	24,264 (97.11)	0.0322
2–3	690 (1.70)	4508 (3.88)	0.1322	589 (2.36)	716 (2.87)	0.0319
>3	6 (0.01)	36 (0.03)	0.0107	5 (0.02)	7 (0.03)	0.0052
Glucagon-like peptide-1 receptor agonists			0.0059			0.0052
No	40,476 (100.00)	116,262 (100.00)		24,987 (100.00)	24,987 (100.00)	
Yes						
Insulins			0.2107			0.0297
No	31,443 (77.68)	99,730 (85.78)		19,659 (78.68)	19,352 (77.45)	
Yes	9033 (22.32)	16,532 (14.22)		5328 (21.32)	5635 (22.55)	
Non-steroidal anti-inflammatory drugs			0.8256			0.0937
No	6349 (15.69)	60,183 (51.76)		6081 (24.34)	5106 (20.43)	
Yes	34,127 (84.31)	56,079 (48.24)		18,906 (75.66)	19,881 (79.57)	
Statins			0.2569			0.0430
No	28,760 (71.05)	95,178 (81.87)		18,623 (74.53)	18,149 (72.63)	
Yes	11,716 (28.95)	21,084 (18.13)		6364 (25.47)	6838 (27.37)	
Aspirin			0.2706			0.0342
No	28,799 (71.15)	95,881 (82.47)		18,291 (73.20)	17,909 (71.67)	
Yes	11,677 (28.85)	20,381 (17.53)		6696 (26.80)	7078 (28.33)	
Duration of T2D (year)	4.09 ± 3.68	2.08 ± 3.13	0.5901	3.44 ± 3.29	3.54 ± 3.70	0.0293
HbA1c check-up > 2 per year			1.3409			0.0466
No	35,846 (88.56)	39,944 (34.36)		20,383 (81.57)	19,923 (79.73)	
Yes	4630 (11.44)	76,318 (65.64)		4604 (18.43)	5064 (20.27)	

^§^: A standardized mean difference of ≤0.1 indicates a negligible difference between the two cohorts. Abbreviation: SD, standard deviation; SMD, standardized mean difference; HbA1c, glycated hemoglobin; T2D, type 2 diabetes.

**Table 2 ijerph-19-11045-t002:** Incidence and HRs of main outcomes associated with metformin users compared with nonusers in patients with T2D.

Variable	Non-Metformin Users			Metformin Users			Crude HR		Adjusted HR ^#^	
	Event	Person-Years	IR *	Event	Person-Years	IR *	HR (95% CI)	*p*-Value	HR (95% CI)	*p*-Value
New-onset chronic urticaria	496	142,255	3.49	819	158,517	5.17	1.49 (1.33, 1.66)	<0.0001	1.56 (1.39, 1.74)	<0.0001
Severe chronic urticaria	9	145,454	0.06	4	164,410	0.02	0.39 (0.12, 1.27)	0.1177	0.40 (0.12, 1.30)	0.1264
Hospitalization for chronic urticaria	19	145,399	0.13	33	164,297	0.2	1.53 (0.87, 2.70)	0.1369	1.45 (0.82, 2.56)	0.2009

^#^: Adjusted HR estimated by the model including the variables of metformin, gender, age, comorbidities, and medications. *: Per 1000 person-years. Abbreviation: CI, confidence interval; HR, hazard ratio; IR, incidence rate; T2D, type 2 diabetes.

**Table 3 ijerph-19-11045-t003:** Incidence and HRs of new-onset chronic urticaria, severe chronic urticaria, and hospitalization for chronic urticaria associated with days’ supply of metformin in T2DM patients.

Variable	Event	Person-Years	IR *	Crude HR		Adjusted HR ^#^	
				HR (95% CI)	*p*-Value	HR (95% CI)	*p*-Value
New-onset chronic urticaria							
Non-metformin users	496	142,255	3.49	1 (Reference)		1 (Reference)	
Metformin users							
1–31 days/years	177	42,992	4.12	1.19 (1.00, 1.41)	0.0515	1.15 (0.97, 1.37)	0.1051
32–88 days/years	164	39,889	4.11	1.18 (0.99, 1.41)	0.0599	1.21 (1.01, 1.44)	0.0346
89–145 days/years	142	39,230	3.62	1.04 (0.86, 1.25)	0.6950	1.17 (0.97, 1.41)	0.1021
>145 days/years	336	36,405	9.23	2.67 (2.32, 3.07)	<0.0001	3.02 (2.62, 3.48)	<0.0001
Severe chronic urticaria							
Non-metformin users	9	145,454	0.06	1 (Reference)		1 (Reference)	
Metformin users							
1–31 days/years	4	164,410	0.07	1.10 (0.30, 4.07)	0.8867	1.15 (0.30, 4.43)	0.8344
32–88 days/years			0.00	NA	NA	NA	NA
89–145 days/years			0.00	NA	NA	NA	NA
>145 days/years			0.03	0.42 (0.05, 3.31)	0.4097	0.42 (0.05, 3.36)	0.4131
Hospitalization for chronic urticaria							
Non-metformin users	19	145,399	0.13	1 (Reference)		1 (Reference)	
Metformin users							
1–31 days/years	9	43,815	0.21	1.57 (0.71, 3.47)	0.2649	1.38 (0.62, 3.08)	0.4274
32–88 days/years	7	41,090	0.17	1.30 (0.55, 3.09)	0.5539	1.25 (0.53, 2.98)	0.6124
89–145 days/years	5	40,744	0.12	0.94 (0.35, 2.52)	0.9009	0.94 (0.35, 2.53)	0.8958
>145 days/years	12	38,648	0.31	2.37 (1.15, 4.89)	0.0191	2.27 (1.09, 4.74)	0.0284

^#^: Adjusted HR estimated by the model including the variables of metformin, gender, age, comorbidities, and medications. *: Per 1000 person-year. Abbreviation: CI, confidence interval; HR, hazard ratio; IR, incidence rate; T2DM, type 2 diabetes mellitus.

## Data Availability

Data of this study are available from the National Health Insurance Research Database (NHIRD) published by Taiwan National Health Insurance (NHI) Administration. The data utilized in this study cannot be made available in the paper, the supplemental files, or in a public repository due to the “Personal Information Protection Act” executed by Taiwan government starting from 2012. Requests for data can be sent as a formal proposal to the NHIRD Office (https://dep.mohw.gov.tw/DOS/cp-2516-3591-113.html, accessed on 8 June 2022) or by email to stsung@mohw.gov.tw.

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
