# Peer review of "Metformin and the Risk of Chronic Urticaria in Patients with Type 2 Diabetes"

_ijerph, 2022, doi:10.3390/ijerph191711045_

Round 1

Reviewer 1 Report

1. Please describe the risk factors for metformin-induced urticaria in the introduction. For example, women, obesity, etc. This would explain why sex, age, obesity, smoking, alcohol-related disorders, comorbidities, and medications were matched between metformin users and non-users.

2. In conclusion, the significance of this study, its application in clinical practice, and future research directions should be presented. Therefore, please move on to the discussion of references 29 and 30 in the conclusion.

Author Response

Thank you for reviewing our manuscript, we reply your comments with the attached word file.   

Reviewer 2 Report

In this work, the authors compare the risk of new onset chronic urticaria in a large cohort of Taiwanese T2D patients. The patients are split into two matched groups, metformin users and non-users. The results indicate strong effects of metformin in chronic urticaria development and severity, and a non-significant effect in hospitalization. From a quick google search, I understand that this study is part of series of works from the same authors studying the effects of metformin in urticaria, asthma, bacteria pneumonia etc for the same cohort of T2D patients.  

I would like to ask the authors to address the comments below:

1.     The introduction discusses the risk of urticaria in T2D patients. There is some weak evidence that metformin could also be associated to skin rash, severe itching, swollen red skin etc (NHS) and urticaria (various case reports). I agree with the sentence “our study is the first to explore the relationship between metformin and chronic urticaria” (lines 225-226) but please include the existing case studies and references in your literature review.

2.     The propensity score and matching procedure followed is clear and the SMDs are defined for each variable of interest (Table 1). I would like to see the post-match overall SMD (a summarized SMD considering all variables entering the model) in order to clearly see how well the two groups are matched.

3.     In Table 1 how do you define obesity, ie what is the BMI cutoff? Is there a reason the authors did not use the BMI itself instead of the factorized obese / not obese which is not so clear (among other in the matching procedure)?

4.     It has been reported previously that the urticaria incidence in females is higher than in males which is also found in this study. It would be interesting to see, separately in males and females, the number of metformin vs no metformin patients developing urticaria and the associated Cox PH estimates. I did not see this this interaction term examined in this study (here only the numbers and estimates in all patients are reports: lines 161-173).

5.     Similar to point 4 above, it would be interesting to examine the incidence rates and statistics in the other significant variables of your study: coronary artery disease, chronic obstructive pulmonary disease, using sulfonylurea, insulin, and non-steroidal anti-inflammatory drugs.

6.     It has been reported that urticaria, asthma and bacterial pneumonia may be related, ie asthma and urticaria are linked and that they frequently occur in the same patient and mycoplasma pneumonia (MP) infection might be pathogenically closely related to urticaria (from another work on Taiwanese patients). I have seen from your previous works these endpoints are present in the dataset. Please use your data to confirm these relationships (and to what degree, ie please report summary statistics).

7.     Please comment if it is possible that urticaria, asthma and bacterial pneumonia are connected in such a way so that one condition (partly) causes the other. If such a relationship is true, it is not clear that the significant effects reported here are strictly due to metformin usage. They could be explained by the development of other conditions (eg asthma, pneumonia) that are not listed in the exclusion criteria or controlled by the model.

8.     In the Discussion section, there is a long paragraph (lines 198-220) discussing the effect of metformin on other endpoints, eg asthma. None of these variables are jointly studied here (although they are clinically significant and should have been) so I do not see the reason to include this in the discussion (the way the manuscript is written the paragraph seems unconnected to the present study).

9.     I would like to ask the authors to make the raw data available to the scientific community (with encrypted patient ID information). This is a very big dataset with a lot of useful information to be extracted and to be studied by other groups interested in the field.

Author Response

Thank you for giving us insightful comments! We reply your comments with the attached Word file. 

Reviewer 3 Report

The study is interesting with a strong methodology. It has minor writing flaws (Figure 1) and a few misspelled words in English.

Introduction: It does have originality. Literature is scarce (1 report showing diabetes mellitus has a slightly higher risk of  chronic urticaria). It offers the potential biological link between the origin of urticaria and metformin. Could you please mention what would be the consequences of chronic urticaria in patients with diabetes?

Material and methods: Strong study design, sample size, and statistical analysis.

There is an error in the wording of the age exclusion criteria in Figure 1. It should be < 20 and > 80, not between 20 and 80.

Results. Clear and understandable.

Discussion. Correct identification of limitations, especially a family history of allergic or autoimmune diseases.

Author Response

Thank you for your encouragement. We reply your comments with the attached Word file.

Reviewer 4 Report

This retrospective cohort study sought to determine the association between metformin use and chronic urticaria in patients with type 2 diabetes. Other than that, I have several observations:

1.      Introduction: although it is recommended that in this section of a manuscript, the review of the literature be sufficient, but not exhaustive, this section leaves me with important doubts about the biological link between the variables analyzed. The most serious of them is the biological plausibility between the use of metformin and the development of urticaria. You mention that (I quote) "Animal studies have demonstrated that metformin can alter the levels of cytokines [7], the ratio of T regulatory (Treg) and T helper 17 (Th17) cells 68 [7,8], and the activation of mast cells [9]”. All of these are anti-inflammatory properties of metformin. So, you should explain in more detail the biological mechanism that links metformin to chronic urticaria.

Suddenly you changed the course of the introduction saying (I quote): “Since most patients have T2D after 50 years [10], 69 chronic urticaria mainly occurs in adults [3,4]". Here I do not find the biological relationship between one thing and another, since, based on the literature reviewed, it is known that metformin has anti-inflammatory activity and that, in fact, it inhibits IgE and the activation of mast cells (reference 9 of your manuscript). Therefore, it is not surprising that this is the first study to look for an association between chronic urticaria (mostly allergic in nature) and metformin (a drug with anti-inflammatory and anti-allergic properties). Such an association does not seem to have plausible biological support.

The most important thing missing from the introduction, then, is the source of your research problem, that is: Are there previous studies between metformin and urticaria (or other autoimmune/allergic diseases)? If not, is there information at the cellular level that would suggest that this drug has properties contrary to those widely known? In other words, why do you thought metformin could be associated to chronic urticaria?

2.      Materials and Methods: In general, it is a well described section. However, an important aspect in the search for the association is the dose of metformin used per day. I recommend that you include this variable in your analysis.

3.      Results: You state that (I quote): “The longer average cumulative duration of metformin use was associated with higher risks of new-onset chronic urticaria and hospitalization for chronic urticaria than metformin no-use (Table 3)”. This statement is incorrect, since, looking at the cited table, metformin users only had a significantly increased risk of new onset chronic urticaria with 32-88 days/years of use (p = 0.0599, aHR 1.21 (1.01, 1.44 [this is borderline significance, by the way]), and at 145 days/years. For chronic severe urticaria and hospitalization there was no significant association. You should correct your statement.

4.      Discussion: In the second paragraph of your discussion, you mention the studies related to metformin and autoimmune diseases. In all the studies you cited, except for a cohort previously published by your research group, metformin is associated with a lower risk of autoimmune diseases and their complications. Here you should give a possible biological explanation for why your results contradict those of other authors, not just emphasize the propensity score matching procedure. Finally, the case reports that you cite at the end of this paragraph are not enough to explain why metformin could be associated with the development of bullous pemphigoid, lichen planus, leukocytoclastic vasculitis, etc.

You correctly mention, in the limitations, that family history of allergic or autoimmune diseases, dietary habits, exercise, occupational exposure, and air pollution are missing. However, I think you should recognize that, despite the strict statistical confounding procedure you did, the lack of that history may have overestimated that association.

5.      Conclusion: this paragraph “Metformin is currently the safest and most frequently used antidiabetic drug world-wide; however, it still has some side effects. Some patients get diarrhea after metformin use. Reports show that long-term metformin use results in vitamin B12 deficiency [29], and metformin use in patients with end-stage renal disease may increase the mortality risk [30]”, does not have to do with the manuscript. You should remove it and focus on results and implications for clinical practice and future research.

Author Response

Thank you for your meaningful recommendations. We reply your comments with the attached Word file.

Round 2

Reviewer 2 Report

Dear authors, thank you for addressing my comments. The manuscript and the analysis are clear now. As an advocate of the open science movement, I would like at some point to be able to access the actual dataset you are analyzing and get a better idea of this valuable information.

In general I am happy with your responses. I am still not sure you replied adequately to:

1. "... see the post-match overall SMD (a summarized SMD considering all variables entering the model) in order to clearly see how well the two groups are matched."

Response: We used nearest neighbor matching to match each treatment group participant with the closest possible untreated group participant, and assumed the standardized mean difference (SMD) of less than 0.1 as negligible difference between the study and control groups. In this study, the SMD of each variable was less than 0.1 after matching (the rightmost column of Table 1). However, we did not take the propensity score as a covariate for matching. Therefore, we could not provide an overall SMD to clearly see how well the two groups are matched. We are so sorry for this inconvenience.

I expected to see some version of an overall SMD for us to understand how the patients are matched when integrating all information. I did not see in your response a clear reason why this was not possible to deliver.

2. "... It would be interesting to see, separately in males and females, the number of metformin vs no metformin patients developing urticaria and the associated Cox PH estimates..."

Response: We have presented these data on supplementary Table S2.

Apologies if I misunderstand Table S2 but I don't see how the statistics depicted there answered this question. I see that the females serve as the reference, thus the males statistics are relative to the reference, i.e. there is a drop in the male hazard relative to the females. I do not see the hazards (depicted in table S2 ) for males and females separately. 

Author Response

We thank you for your encouragement. We have responsed to your comments on the attahced file.   

Reviewer 4 Report

Thank you for attending my comments. Good job!

Author Response

Thank you for your warm encouragement!